# New Class of Benzodiazepinone Derivatives as Pro-Death Agents Targeting BIR Domains in Cancer Cells

**DOI:** 10.3390/molecules28010446

**Published:** 2023-01-03

**Authors:** Michele Fiore, Michele Mosconi, Francesco Bonì, Alice Parodi, Annalisa Salis, Bruno Tasso, Eloise Mastrangelo, Enrico Millo, Federica Cossu

**Affiliations:** 1National Research Council (IBF-CNR) Genoa Unit, Institute of Biophysics, Via De Marini 6, 16149 Genova, Italy; 2National Research Council (IBF-CNR) Milan Unit, Institute of Biophysics, Via Celoria 26, 20133 Milan, Italy; 3Department of Experimental Medicine, Section of Biochemistry, University of Genoa, Viale Benedetto XV 1, 16132 Genova, Italy; 4Department of Pharmacy, Section of Medicinal Chemistry, School of Medical and Pharmaceutical Sciences, University of Genoa, Viale Benedetto XV 3, 16132 Genova, Italy

**Keywords:** baculoviral IAP repeat (BIR), inhibitor of apoptosis protein (IAP), NF-κB (nuclear factor kappa light chain enhancer of activated B cells), benzodiazepinone, virtual docking, MDA-MB-231

## Abstract

Inhibitor of Apoptosis Proteins (IAPs) are validated targets for cancer therapy, and the deregulation of their activities within the NF-κB pathway correlates with chemoresistance events, even after treatment with IAPs-antagonists in the clinic (Smac-mimetics). The molecule FC2 was identified as a NF-κB pathway modulator in MDA-MB-231 adenocarcinoma cancer cells after virtual screening of the Chembridge library against the Baculoviral IAP Repeat 1 (BIR1) domain of cIAP2 and XIAP. An improved cytotoxic effect is observed when FC2 is combined with Smac-mimetics or with the cytokine Tumor Necrosis Factor (TNF). Here, we propose a library of 22 derivatives of FC2, whose scaffold was rationally modified starting from the position identified as R_1_. The cytotoxic effect of FC2 derivatives was evaluated in MDA-MB-231 and binding to the cIAP2- and XIAP-BIR1 domains was assessed in fluorescence-based techniques and virtual docking. Among 22 derivatives, **4m** and **4p** display improved efficacy/potency in MDA-MB-231 cells and low micromolar binding affinity vs the target proteins. Two additional candidates (**4b** and **4u**) display promising cytotoxic effects in combination with TNF, suggesting the connection between this class of molecules and the NF-κB pathway. These results provide the rationale for further FC2 modifications and the design of novel IAP-targeting candidates supporting known therapies.

## 1. Introduction

Benzodiazepinones belong to the class of 1,4-benzodiazepins currently used in clinics as treatments for muscle spasms, anxiety, depression and insomnia [1]. Diazepam (1,4-benzodiazepin-2-one) is one of the best-known drugs of this class, presenting anxiolytic effects triggered by an increased inhibitory activity of Gamma-aminobutyric acid (GABA) [2]. Over and above the standard clinical uses of benzodiazepinones, different studies have demonstrated a wide variety of activities; among others, antiviral, herbicidal and neuro-protective activities emerge as the most relevant ones [3,4]. Potential anti-cancer activities have been recently discovered and different benzodiazepinone-like molecules are currently under analysis as potential treatments for many types of cancer. A series of 1,4-benzodiazepin-2,5-dione derivatives have been shown to perturb the interaction between p53 and Hdm2 reducing cell viability in wild-type p53 tumors, raising p53 levels and inducing cell growth arrest [5]. Moreover, clinical studies on the benzodiazepinone TLN-4601 for the treatment of patients with glioblastoma have demonstrated a dual effect of this molecule, simultaneously binding peripheral benzodiazepine receptors (PBR) and inhibiting the Ras-MAPK pathway [6,7]. Since PBR could act as an endogenous modulator of mitochondrial apoptosis and increased PBR levels have been detected in many tumors, benzodiazepinone-like molecules could induce apoptosis interacting with PBR (with affinities in the low micromolar range) in tumor cells promoting Mitochondrial Outer Membrane Permeabilization (MOMP), an irreversible step in the apoptotic pathway [8,9,10,11]. An antagonistic activity of benzodiazepinone-like molecules against X-linked Inhibitor of Apoptosis Protein (XIAP) was reported, eliciting the activity of caspases in xenografted pharmacodynamic models [12,13]. This mechanism leads to apoptosis and cell viability reduction via the direct inhibition of the type II BIR (Baculoviral IAP Repeat) domain of XIAP. More recently, a virtual screening against type I BIR domains of Inhibitor of Apoptosis Proteins (IAPs) [14] identified FC2 as a benzodiazepinone-like molecule capable to modulate the NF-κB pathway and to induce apoptosis in MDA-MB-231 cells targeting both XIAP and cellular IAP (cIAP) type I BIR domains. Inhibitor of Apoptosis Proteins are overexpressed in many tumors and this correlates with cancer onset, progression and poor prognosis. The study of the molecular bases of IAPs function in tumors has revealed the important role of type I BIRs (of cIAP2 and XIAP) in the recruitment of IAPs to pro-survival macromolecular complexes with TRAF1/2 [15] and TAB1 [15,16]. Such interactions are essential for the regulation of canonical and non-canonical NF-κB pathways, participating in inflammation, death/survival balance and, importantly, in chemoresistance events observed upon treatment with known IAP antagonists [17,18]. For this reason, type I BIRs (BIR1) were targeted using a structure-based approach to find molecules able to modulate the assembly of BIR1 domains with their partner proteins. FC2 was shown to enforce the pro-death effect of known IAP antagonists and, at certain concentrations, to synergize with the cytokine Tumor Necrosis Factor (TNF), which activates NF-κB. FC2 modulates the interaction between XIAP and TAB1, leading to the prolonged activation of NF-κB, as demonstrated by delayed recovery of IκBα levels and alteration of phosphorylation patterns of specific NF-κB markers (p65, TAK1, MKK3/6).

The FC2 chemical structure includes a benzodiazepinone core as a scaffold, decorated with an indolic group and a benzoyl group on opposite sides (Figure 1, Table 1). Compared to other compounds sharing the benzodiazepinone, the position of the indolic group and the presence of the benzoyl group confer unique features of the benzodiazepinones reported as antagonists of IAPs. In preliminary Structure–Activity Relationship (SAR) experiments, the deletion of the benzoyl group induced a re-orientation of the compounds’ predicted poses in the α1-turn-α2 region of type I BIRs and the loss of the cytotoxic activity (analogs 1 or 7 [14]). In this work, we started the rational modification of FC2 designing a library of 22 derivatives and substituting the indolic group in position R_1_ of the central scaffold. The library was screened in human breast adenocarcinoma cells (MDA-MB-231) and cell viability data were combined with (i) the structure-based analysis of the predicted poses of FC2 derivatives on type I BIRs and (ii) their effects/affinity measured in vitro on the purified cIAP2- and XIAP-BIR1 domains.

## 2. Results

### 2.1. Chemistry

The synthesis procedure used for the synthesis of the FC2 derivatives (Table 1) was adapted from previous protocols [19,20,21]. The reactions were carried out by a condensation reaction between 3,4-diaminobenzophenone **1** and 5,5-dimethylcyclohexane-1,3-dione **2a** or cyclohexane-1,3-di-ones **2b** in H_2_O–EtOH (1:1) at 80 °C for about three hours (Figure 1). After nearly complete conversion into the corresponding enamines as major regioisomers **3a** or **3b**, **3a** and **3b** were separated by their isomers by preparative HPLC and the intermediates were then condensed with appropriate aldehydes in the presence of catalytic amounts of acetic acid to furnish 1H-dibenzo[1,4]diazepin-1-one derivatives **4a**–**4v**. The crude products were crystallized or purified with HPLC preparative to obtain the final products in moderate yield. The mechanism hypothesis for the formation of 1H-dibenzo[1,4]diazepin-1-one derivatives is summarized in Appendix A.

### 2.2. SAR of FC2 Derivatives on MDA-MB-231 Human Breast Adenocarcinoma

#### 2.2.1. Compound **4d** As a Novel Base for the Design of New Putative NF-κB Modulators

The screening of FC2-like molecules initiated by Cossu et al. [14] was performed in human breast adenocarcinoma cells (MDA-MB-231). Such a cell line is extensively investigated in the presence of IAP-targeting compounds and was used to characterize the mechanism of FC2-induced cell death and the regulation of NF-κB [14]. For this reason, MDA-MB-231 cells were adopted to continue the screening of novel FC2 derivatives (Table 1). Preliminary Structure-Activity Relationship (SAR) analysis suggested that the addition of bulky moieties in R compromise the pro-death activity of compounds [14]. To corroborate these results, starting from FC2 (EC_50_ = 11.38 μM) we inserted a gem-dimethyl moiety at position 5 of the cyclohexane-1,3-dione (**4a**), which was shown to negatively affect the potency (EC_50_ = 42.76 μM). Based on these initial, promising findings, compound FC2 was selected as the reference molecule to explore the SAR around this chemotype, keeping the same substituent of FC2 (hydrogen) in R, the central scaffold (7-benzoyl-2,3,4,5,10,11-hexahydro- 1H-dibenzo[b,e][1,4]diazepin-1-one) moiety unmodified and focusing the SAR study on position R_1_ (Figure 1).

The indole substitution by a piperidine with nitrogen in para reduces the activity that is completely lost when the piperidine’s nitrogen is in meta [14]. We then explored the role of the heteroaryl group in R_1_ by replacing the 1H-indole with other rings, such as thiophene (**4b**) or 2-methylfuran (**4c**). Compound **4b** retains a good effect on the MDA-MB-231 cell line (EC_50_ = 21.9 μM), whereas the apolar nature of R_1_ in **4c** causes a marked drop in potency (EC_50_ = 65 μM). To further explore the chemical space around R_1_, we considered substituting heterocycles with a phenyl ring (**4d**). Such a group allows the evaluation of a variety of substitutions in different positions of the ring. 

#### 2.2.2. Identification of the Best Mono-Substitutions of R_1_

Starting from **4d** (EC_50_ = 19.57 μM), several phenyl-substituted analogues (**4e**–**4v**, Table 1) were synthesized. First, the effect of the para substitution on the phenyl ring was investigated with the addition of electron-donating groups: alkyl (**4e** EC_50_ = 38.87 μM), alkyl amino (**4f**, EC_50_ = 39.15 μM), alkyloxy (**4g** EC_50_ = 67.23 μM), or hydroxyl (**4j** EC_50_ = 48.18 μM) that caused a progressive decrease in activity. The addition of electron-withdrawing groups such as trifluoroalkyl **4h** or halogen **4i** did not show improvements in cytotoxicity compared to **4d** (EC_50_ = 22.32 μM and EC_50_ = 25.89 μM, for **4h** and **4i**, respectively). According to the structural analysis, moving R_1_ substitutions from the para to the meta position could improve R_1_ accommodation. In fact, a significant increase in activity is observed with the methoxy group in meta (**4k**) from the para (**4g**) (EC_50_ = 10.26 μM vs. EC_50_ = 67.23 μM). The same trend was observed with the hydroxyl group (in meta **4l** EC_50_ = 18.58 μM vs in para **4j**, EC_50_ = 48.18 μM). To deepen the contribution of the meta position on the activity, we synthesized a derivative containing the nitro group (**4m**), whose cytotoxicity (EC_50_ = 5.54 μM) overcomes the one observed for FC2, being one of the most potent analogues within this small set. On the contrary, the reduction of the same group to the aminic one (**4n**) causes a loss of activity. 

#### 2.2.3. Di- and Tri-substituted Derivatives

As a further step in the exploration of SAR within this chemotype, di- and tri-substituted phenyl derivatives were also explored. Given the encouraging results with the methoxy and hydroxyl groups, such moieties were introduced both in para and meta positions (**4o** EC_50_ = 13.47 μM; **4p** EC_50_ = 6.92 μM). More hindered groups, such as 2,2-difluorobenzo[d][1,3]dioxole (**4q**), displayed a comparable cytotoxic effect (EC_50_ = 14.68 μM), being the hydroxyl group preferred in these positions of the ring. Since the introduction of a second substituent on the phenyl ring proved in general to be beneficial, leading to some compounds with efficacy comparable to that of the progenitor, we maintained the methoxy group in meta inserting a hydroxyl group in para. The new compound, called **4r**, produces one of the most active compounds of this series with EC_50_ = 7.49 μM. The same structure with the inversion of the substituents (i.e., hydroxyl in meta and methoxy in para), **4s** moderately lowers the activity (EC_50_ = 15.25 μM). Keeping the methoxy group in meta position and moving the hydroxyl group from the para to the ortho position, **4t** results in a molecule without activity highlighting how the replacement in the ortho is not optimal to carrying out activities. To conclude the screening of the preferred positions of most cytotoxic R_1_ substitutions, we combined the hydroxyl moiety in para with the nitro in meta. However, an additive cytotoxic effect with **4u** was not observed (EC_50_ = 17.03 μM), nor with a tri-substituted ring using the combination of the three substituents in their preferred positions. Indeed, **4v** shows weak activity (EC_50_ = 31.45 μM).

### 2.3. Cytotoxic Effect of the Derivatives in Combination with TNF

Since the cytokine TNF stimulates the TNF-R and the recruitment of all molecular actors (i.e., IAPs) triggering the NF-κB pathway, the effect of each compound on cell viability was measured both as a single agent and in combination with 50 ng/mL of TNF (Table 1). The experiments were carried as reported by Cossu et al. [14], where the induction of cell death with FC2 was shown to be potentiated by co-treatment with TNF. The derivatives **4a**, **4c**, **4f**, **4g** displayed enhanced activity in combination with TNF and, in particular, **4b** and **4u** reached EC_50_ values comparable to the ones observed for FC2 (EC_50_ = 9.20 μM and 9.53 μM, respectively). Concerning **4i**, **4k**–**4t**, **4v**, the combination with TNF does not seem to affect the cytotoxicity registered as single agents, **4m**, **4p** and **4r** being the best FC2 derivatives, both in mono treatment and in combination with TNF in cell viability assays. Interestingly, TNF combination with **4d**, **4e**, **4h**, **4j** seems to provide a protective effect. These results show that this class of compounds is worthy of being further investigated in the context of NF-κB regulation, since the variability of responses upon different R_1_ modifications could reveal details about their mechanism of action.

### 2.4. In Silico Characterization of FC2 Derivatives Binding to Type I BIRs

The FC2 compound was identified in virtual screening against the type I BIR domains of IAPs. In particular, the α1-turn-α2 region of BIR1 was set as a target surface, since it is involved in protein–protein interactions (PPI) crucial for cancer cell survival. Virtual docking simulations of FC2 derivatives performed against the whole surface of BIR1 domains confirmed that FC2 derivatives preferentially target the same hotspot region (the α1-turn-α2 region) predicted for FC2 [14] (Figure 1a), suggesting their ability to perturb the assembly of BIR1-mediated complexes between cIAP2/TRAF2 and XIAP/TAB1. The FC2 derivatives also show free energies of binding comparable to FC2, with estimated inhibition constants (Ki) in the low micromolar range (Table 2). Major deviations were calculated for **4h** in cIAP2-BIR1 and for **4g** in XIAP, whose predicted Ki reached 9.09 μM and 4.83 μM, respectively. Virtual docking simulations allowed the description of the derivatives’ interactions with the target proteins, as well as to hypothesize the correlations between some predicted interactions and observed enhancements in cytotoxicity.

#### 2.4.1. The R_1_ Modification Is Relevant to the Positioning of FC2 Derivatives in the Structural Hotspot

Concerning predicted poses, R_1_ modification influenced the localization of the central scaffold and the orientation of the derivatives, thus influencing the contacts of the interaction networks, compared to the typical FC2 pose (Figure 1a). 

*cIAP2-BIR1*—Compared to XIAP, E47 in cIAP2 provides a polar environment, shrinking the cavity hosting the central scaffold and R_1_. As a result, the bulky groups in R_1_ induced significant repositioning of the derivatives compared to FC2. Less significant effects were predicted for **4j**, **4n**, **4s**, **4u,** where the typical orientation of FC2 was maintained but shifted towards α1. In these cases, the R_1_ groups established interactions with the turn between α1 and α2, losing however the typical FC2 positioning of the indole. Compounds **4a**, **4b**, **4d**, **4k**, **4m**, **4o**, **4p**, **4r** maintained the typical FC2 pose in the hotspot.

*XIAP-BIR1*—Compared to FC2, the poses of some derivatives in XIAP were inverted, being the localization of the scaffold maintained in front of the hydrophobic cleft delineated by α1-α2 interface. Notably, this inversion occurred upon the modification of R_1_ with 2-methylfuran, as in **4c** or bulky groups in the para position of the phenyl ring, as the methoxy group found in **4g**, or with the di- or tri-substituted phenyl groups in R_1_. Other compounds were predicted to lose the localization of the scaffold, slipping away from the hydrophobic cleft. Such an effect is accompanied or driven by novel interactions established by R_1_ with the residues of the turn region, as for **4u**, or with residues outside the hotspot, as for **4s**, **4v**. Derivatives **4b**, **4d**, **4f**, **4h**, **4i**, **4k**, **4m**, **4o**, **4p**, **4q**, **4r**, **4t** are shown to maintain the typical FC2 pose.

#### 2.4.2. Cytotoxic FC2 Derivatives Retain FC2 Pose and Interaction Network in the BIR1 Structural Hotspot

The target region of type I BIRs comprises α1-turn-α2 secondary structures, whose packing is stabilized by hydrophobic/electrostatic interactions between conserved residues (Figure 1). In this region, the conserved residues are responsible for critical contacts stabilizing the whole BIR fold, as previously described [25]. 

Comparing the poses of FC2 derivatives and experimental data, the retention of the typical FC2 pose generally correlates with the preservation of the cytotoxic effect in MDA-MB-231. Furthermore, less active compounds in cell viability assays induce the destabilization of type I BIRs folding, as suggested by negative ΔT_M_ reported in 2.5.1 and Table 2. For this reason, the analysis of the poses predicted for FC2 and derivatives focused on contacts between the different portions of the compounds (Figure 2a) with the residues relevant for the BIR folding (Figure 1c). 

*Benzoyl and scaffold*—The helices’ packing is stabilized by the hydrophobic core, constituted of bouncing hydrophobic/van der Waals interactions between a conserved (or conservatively substituted) set of residues (XIAP F33, L47, F36 and V42, corresponding to cIAP2 Y36, L50, F39 and V45, respectively). The central scaffold of FC2 and of the cytotoxic FC2 derivatives (Figure 2a) faces this hydrophobic cleft in XIAP, thus allowing positioning of the benzoyl group in the proximity of a helix groove between M33/L30 and Y36/F33 from the α1 of cIAP2 and XIAP, respectively (Figure 2b,c). In cIAP2, the hydrophobic cleft is interrupted by the presence of E47 (A44 in XIAP), which engages hydrogen bonds with the amine group of the central scaffold. Non-conservative substitutions (XIAP S40 > cIAP2 V45; XIAP A44 > cIAP2 E47) induce a different positioning of the scaffold of FC2 and the cytotoxic derivatives in XIAP and cIAP2 (Figure 1a,b). 

*R_1_ substitution*—The region including the turn and the first residues of α2 hosts the R_1_ substitution. Notably, two conserved proline residues act as cornerstones or hinges at the borders of the turn: P37/P40 terminate α1, whereas P41/P44 are located before α2 in XIAP and cIAP2, respectively. These two proline residues confer a particular bend to the turn, which allows the accommodation of the FC2 indole and of the R_1_ of the cytotoxic FC2 derivatives (Figure 2). Such an accommodation is stabilized by interactions between R_1_ and the backbone of F39-S46/F36-S43 in cIAP2/XIAP. 

#### 2.4.3. Electron Withdrawing Groups in the Meta Position of R_1_ Improve the Interaction Network

Among the derivatives maintaining the FC2 pose, the **4m** overall energetic profile is improved compared to FC2. Indeed, the candidate **4m** carries an electron withdrawing group in R_1_ (the nitro group in meta of the phenyl ring) that gains hydrogen bonds with the backbone of G42 and V43 of cIAP2 (Figure 2b). The addition of electron withdrawing groups to the phenyl ring in R_1_ is predicted to be successful also in the case of candidate **4p**. Compared to FC2, the presence of two hydroxyl groups (one in meta and one in para) is shown to engage novel hydrogen bonds with a P40 backbone and E47 side chain and electrostatic interactions with F39 and V43 backbones in cIAP2 (Figure 2b). Analogously to the nitro group of **4m**, the hydroxyl group in meta gains an electrostatic interaction with S38 in XIAP (Figure 2c) that was not observed for the indole of FC2.

### 2.5. Effect of FC2 Derivatives on Type I BIRs of cIAP2 and XIAP

#### 2.5.1. Thermal Stability

The melting temperatures of type I BIRs were evaluated in the absence (cIAP2-BIR1 T_M_ 66.7 ± 0.3 °C; XIAP-BIR1 T_M_ 63.6 ± 0.4 °C) or the presence of the derivatives. A set of compounds did not induce a significant variation of the thermal stability, with a ΔT_M_ (T_M BIR1-DERIVATIVE_–T_M BIR1_) ranging from −2.90 to 0.75 °C. Conversely, a set of compounds significantly destabilized the proteins, decreasing the melting temperatures of 6–20 °C. Such observations could depend on a balance between stabilizing and destabilizing interactions engaged by the derivatives with the purified BIR1 domains. Virtual docking suggests that most destabilizing compounds are prone to lose the typical FC2 pose. It could be hypothesized that the R_1_ of the destabilizing compounds intercepts structural elements critical for BIR1 folding. Of note, the derivatives significantly decreasing thermal stability display worse cytotoxic profiles. 

#### 2.5.2. Measurement of FC2 Derivatives’ Affinity for Type I BIRs

The binding of FC2 derivatives to the BIR1 domains was assessed by in vitro fluorimetric assays. Increasing concentrations of FC2 derivatives were used to titrate wild type cIAP2- or XIAP-BIR1 at 23 °C. The decrease in the BIR1 fluorescence was plotted against increasing FC2 derivative concentrations, thus yielding the dissociation constants (Kd) reported in Table 2. Some R_1_ substituents (**4c**, **4e**, **4i**, **4k**, **4s**, **4t**) impair the binding to the purified proteins. In the case of **4f**, **4n**, **4o**, **4p**, **4v,** the affinity is comparable among homologous BIRs, with **4o** and **4q** maintaining a moderate affinity for cIAP2- and XIAP-BIR1. Different Kd values versus cIAP2- or XIAP-BIR1 (3- to 5-fold) observed for FC2, **4a**, **4d**, **4g**, **4h**, **4j**, **4m** and **4q** could be due to the diverse interactions that the derivatives establish with non-conserved protein residues. Such differences could be exploited in the following phases of the design of these candidates, with the aim to develop selectivity against distinct IAP homologues. Notably, some of the most cytotoxic compounds retain affinity for cIAP2- and XIAP-BIR1 in the low micromolar range, as in the case of **4m**, **4o**, **4p**.

## 3. Discussion

In this work, we rationally modified the candidate FC2, identified after a virtual screening of compounds against type I BIR domains of cIAP2 and XIAP [14]. FC2 was shown to modulate the NF-κB pathway and induce cancer cell death in two human breast adenocarcinomas and in ovarian carcinoma cell lines, both as a single agent and in combination with known IAP antagonists or with the cytokine TNF. 

The screening of FC2 analogues started in 2021 [14] on MDA-MB-231, a human breast adenocarcinoma cell line, which was used in this work to assess the cytotoxic effect of newly synthesized FC2 derivatives differently substituted in R_1_. Structure–Activity Relationship studies in breast adenocarcinoma led to the identification of **4m**, **4p** and **4r** with promising cytotoxic effects as single agents. Interestingly, the first two display lower EC_50_ values when administered in combination with TNF. For this reason, according to cell-based studies **4m** and **4p** can be considered the best hits from the library.

Variable effects were observed during the co-treatment of MDA-MB-231 with the cytokine TNF and different derivatives. Some derivatives in combination with TNF display improved cytotoxicity, as in the case of **4b** and **4u**. Conversely, the combination with TNF seems to provide a protective effect in a few cases. Despite such variability, the data suggest interplaying between these derivatives and NF-κB triggering by TNF. For this reason, we plan to further investigate this combination in other cell lines or explore different experimental conditions. 

Type I BIRs lack a particular cavity that type II BIRs use for direct apoptosis inhibition. Conversely, type I BIRs recruit IAP molecular partners using the α1-turn-α2 surface. The structural analysis of this hotspot revealed the interaction network stabilizing the α1-turn-α2 folding involving crucial residues, some of which are predicted to engage contacts with FC2 and derivatives. Analyzing the poses of different portions of the FC2 chemotype, particular positions in the hotspot were confirmed as critical residues to be targeted. In detail, M33/L30 and Y36/F33 of cIAP2/XIAP establish contacts with the benzoyl group of the compounds. Moreover, the hydrophobic core of the hotspot faces the derivatives’ central scaffold, whereas residues F39-S46 in cIAP2, F36-S43 in XIAP, belonging to the turn region accommodate the substitutions in R_1_.

Comparing in silico and in vitro data, it can be noticed that the loss of the typical FC2 pose both in XIAP and in cIAP2 correlates with the impaired cytotoxicity of the derivatives in MDA-MB-231. Meanwhile, the derivatives, whose poses are superimposable to FC2, keep cytotoxic activity. In particular, **4m** and **4p** carry electron-withdrawing substitutions in the meta position of the phenyl ring allowing the optimal accommodation of the R_1_ and gaining novel interactions in the interaction network.

During the biophysical characterization, we observed a set of compounds inducing a significant decrease in the melting temperatures of the isolated BIR1 domains of cIAP2 and XIAP, probably due to engaging interactions in protein regions critical for the folding. Interestingly, these compounds display low cytotoxicity in cancer cell viability assays. According to experimental results, FC2 retains the best measured Kd. Among novel FC2 derivatives, compounds **4a**, **4d**, **4g**, **4h**, **4j**, **4m**, **4o**, **4p**, **4q** are the best binders, displaying Kd values in the low micromolar range. Such results generally confirm the beneficial effects of electron-withdrawing groups’ addition in R_1_ in the meta position of the phenyl ring. Since these candidates are proposed as NF-κB modulators whose regulation depends on the conformation adopted by IAPs, we also intend to explore the effect of such derivatives on the full-length proteins, broadening the study on wider effects on the recruitment of macromolecular complexes.

The combination of virtual and experimental approaches here reported allowed the selection of chemical groups (NO_2_ in meta, or OH in the para and in the meta of the phenyl ring) to be maintained in R_1_, as in **4m**, **4p**, which displayed improved cytotoxic profiles and optimal biophysical properties in terms of affinity for the target proteins and the absence of destabilizing effects. The combination of molecular dynamics and other experimental structure-based techniques (i.e., crystallography, NMR) are in progress to better characterize the molecular details of BIR1 binding with selected compounds **4m** and **4p**. Further modifications will concern the decoration of the benzoyl group, to verify improvements in efficacy/potency within viability assays. The best candidates will be selected for deeper investigations into the cellular mechanisms at the base of their pro-apoptotic effect.

The information gained so far enforces the path to the design of a new class of compounds directed at NF-κB pathway modulation and for the development of new therapeutic strategies hampering cancer cell survival and IAP-dependent chemoresistance.

## 4. Materials and Methods

### 4.1. Chemistry

#### 4.1.1. Experimental Instrumentation

All solvents and chemicals were reagent grade. Unless otherwise mentioned, all solvents and chemicals were acquired from Sigma Aldrich (St. Louis, MO, USA), VWR (Radnor Township, PA, USA) and used as received unless purified.

A rotary evaporator allowed solvent removal at ca.10–50 Torr. The analytical instrument used was Agilent 1260 high performance liquid chromatography (HPLC, Agilent Technologies, Santa Clara, CA, USA). The analytical HPLC column was a Phenomenex (Torrance, CA, USA) C18 Luna (4.6 × 250 mm, 5 μm).

The preparative HPLC was Agilent 1260 Infinity preparative HPLC and the column Phenomenex C18 Luna (21.2 × 250 mm, 15 μm) was used for preparative chromatography. Liquid chromatography-electrospray mass spectrometry (HPLC-ESI-MS, Agilent Technologies, Santa Clara, CA, USA) was used to analyze the intermediates and the raw products with an Agilent 1100 series LC/MSD ion trap instrument.

The HRMS experiments were performed using Q Exactive Orbitrap instrument by Thermo Scientific (Waltham, MA, USA).

The nuclear magnetic resonance (NMR) analyses were performed using a Jeol spectrometer 400 MHz (Jeol LTD, Akishima, Tokyo, Japan).

The proton spectra and the carbon spectra were acquired at room temperature, at 400 MHz and at 100 MHz, respectively. Chemical shifts were reported in δ units (ppm) relative to TMS as an internal standard. Coupling constants (J) were reported in Hertz (Hz).

All the raw products were purified with preparative HPLC using the following gradient: from 0 to 5 min at 20% eluent B, from 5 min to 40 min at 100% eluent B, from 40 to 45 min at 100% eluent B. Eluent A was water with 0.1% formic acid (FOA) and eluent B was acetonitrile with 0.1% FOA. All the final products utilized in biological assays were judged to have a purity of 95% or higher, based on analytical HPLC/MS analysis.

Compound purity was determined by integrating peak areas of the chromatogram obtained in liquid phase, monitored at 254 nm.

#### 4.1.2. Synthesis of FC2 and FC2 Derivatives (**4a-e**)

##### 7-benzoyl-11-(1H-indol-3-yl)-2,3,4,5,10,11-hexahydro-1H-dibenzo[b,e][1,4]diazepin-1-one (FC2)

To a solution of 3,4-diaminobenzophenone (212.2 mg, 1 mmol) in a mixture of water and EtOH, 1,3 cyclohexanedione (112.1 mg, 1 mmol) was added. The reaction was stirred for 3 h at T = 80 °C. The desired product was separated by its isomer by preparative HPLC. The peak of interest was concentrated and lyophilized to obtain 3-((2-amino-5-benzoylphenyl)amino)cyclohex-2-en-1-one as a brown oil (199.1 mg, 65%).

3-((2-amino-5-benzoylphenyl)amino)cyclohex-2-en-1-one (36.8 mg, 0.12 mmol) was dissolved in EtOH and the indole-3-carboxaldehyde (17.4 mg, 0.12 mmol) was added to the mixture; then, a few drops of acetic acid were added to the reaction and the mixture was stirred for 24 h at room temperature. The mixture was centrifuged and the precipitate was washed three times with EtOH and then dried to obtain the final product as a yellow powder (10.4 mg, 20%).

^1^H NMR (400 MHz, DMSO-d_6_) δ 10.74 (s, 1H, NH indole); 9.06 (s, 1H, NH); 7.78–7.37 (m, 7H, arom); 7.21 (s, 1H, arom); 7.17–7.04 (m, 4H, arom); 6.86 (d, J = 6.8, 1H, arom); 6.71 (d, J = 6.4, 1H, CH); 5.74 (d, J = 6.0, 1H, NH); 2.77–2.54 (m, 2H, CH_2_); 2.31–2.13 (m, 2H, CH_2_); 2.01–1.82 (m, 2H, CH_2_).

^13^C NMR (100 MHz, DMSO-d_6_) δ 194.3, 193.1, 145.8, 141.3, 138.9, 136.6, 134.5, 132.2, 130.1, 129.4, 128.8, 127.4, 127.1, 125.7, 122.4, 121.8, 120.3, 118.9, 114.5, 113.1, 112.2, 110.1, 107.2, 55.1, 36.1, 31.1, 21.8.

The HRMS (ESI) was calculated for C_28_H_24_N_3_O_2_ [M + H]^+^ 434.18684; found 434.18599.

##### 7-benzoyl-11-(1H-indol-3-yl)-3,3-dimethyl-2,3,4,5,10,11-hexahydro-1H-dibenzo[b,e][1,4]diazepin-1-one (**4a**)

5,5-dimethyl-1,3-cyclohexanedione (70 mg, 0.5 mmol) and 3,4-diaminobenzophenone (106.1 mg, 0.5mmol) were dissolved in a mixture of water and EtOH. The reaction was stirred for 3 h at T = 80 °C and then overnight at room temperature. The product was purified by preparative HPLC and lyophilized to obtain a brown oil (100 mg, 60%).

3-((2-amino-5-benzoylphenyl)amino)-5,5-dimethylcyclohex-2-en-1-one (40.1 mg, 0.12 mmol) was dissolved in EtOH and indole-3-carboxaldehyde (17.4 mg, 0.12 mmol) was added to the mixture; then, a few drops of acetic acid were added to the reaction and the mixture was stirred for 24 h at room temperature.

The solvent was removed by rotavapor; the mixture was dissolved in acetonitrile and purified by preparative HPLC. The final compound was lyophilized to obtain the compound as a yellow powder (8.9 mg, 16%).

^1^H NMR (400 MHz, DMSO-d_6_) δ 10.86 (s, 1H, NH indole); 9.03 (s, 1H, NH); 7.74–7.33 (m, 7H, arom); 7.19 (s, 1H, arom); 7.13–7.02 (m, 4H, arom); 6.92 (d, J = 8.0, 1H, arom); 6.68 (d, J = 6.0, 1H, CH); 5.83 (d, J = 6.0, 1H, NH); 2.31–2.13 (m, 2H, CH_2_); 2.01–1.82 (m, 2H, CH_2_); 1.04 (s, 3H, CH_3_); 0.97 (s, 3H, CH_3_).

^13^C NMR (100 MHz, DMSO-d_6_) δ 194.3, 193.0, 145.6, 141.3, 138.8, 136.7, 134.5, 132.5, 130.1, 129.4, 128.8, 127.4, 127.0, 125.7, 122.3, 121.9, 120.3, 119.1, 114.5, 113.0, 112.2, 110.1, 107.8, 55.4, 35.8, 31.1, 27.4, 27.1, 21.8.

The HRMS (ESI) was calculated for C_30_H_28_N_3_O_2_ [M + H]^+^ 462.21814; found 462.21798.

##### 7-benzoyl-11-(thiophen-2-yl)-2,3,4,5,10,11-hexahydro-1H-dibenzo[b,e][1,4]diazepin-1-one (**4b**)

Compound **4b** (9.4 mg, 24%) was prepared from thiophene-2-carbaldehyde (11 mL, 0.1 mmol) and 3-((2-amino-5-benzoylphenyl)amino)cyclohex-2-en-1-one (30.5 mg, 0.1 mmol) in the same manner as described for **FC2**.

Then, the final compound was purified by preparative HPLC to obtain the title compound as a yellow oil.

^1^H NMR (400 MHz, DMSO-d6) δ 8.96 (s, 1H, NH); 7.72–7.41 (m, 6H, arom); 7.13 (dd, J = 2.0, 8.4, 1H, arom); 7.07 (dd, J = 1.6, 8.0, 1H, arom); 6.93–6.79 (m, 3H, arom); 6.61 (d, J = 6.0, 1H, CH); 5.70 (d, J = 6.4, 1H, NH); 2.76–2.53 (m, 2H, CH2); 2.29–2.14 (m, 2H, CH_2_); 2.01–1.80 (m, 2H, CH_2_).

^13^C NMR (100 MHz, DMSO-d_6_) δ 194.3, 193.0, 148.4, 141.6, 138.8, 136.7, 132.0, 130.3, 129.5, 128.8, 126.7, 125.6, 122.8, 120.0, 118.4, 112.5, 55.1, 36.4, 31.1, 21.8.

The HRMS (ESI) was calculated for C_24_H_21_N_2_O_2_S [M + H]^+^ 401.13326; found 401.13145.

##### 7-benzoyl-11-(5-methylfuran-2-yl)-2,3,4,5,10,11-hexahydro-1H-dibenzo[b,e][1,4]diazepin-1-one (**4c**)

Compound **4c** (9 mg, 22%) was prepared from 4-methylcyclopenta-1,3-diene-1-carbaldehyde (12 mL, 0.1 mmol) and 3-((2-amino-5-benzoylphenyl)amino)cyclohex-2-en-1-one (30.5 mg, 0.1 mmol) in the same manner as described for **FC2**.

Then, the final compound was purified by preparative HPLC to obtain the title compound as a yellow oil.

^1^H NMR (400 MHz, DMSO-d_6_) δ 9.01 (s, 1H, NH); 7.67–7.38 (m, 6H, arom); 7.11 (dd, J = 2.0, 8.4, 1H, arom); 7.06 (dd, J = 1.6, 8.0, 1H, arom); 6.74 (d, J = 6.4, 1H, arom); 6.59 (d, J = 6.0, 1H, CH); 6.39 (d, J = 6.4, 1H, arom); 5.69 (d, J = 6.0, 1H, NH); 2.75–2.55 (m, 2H, CH_2_); 2.62 (s, 3H, CH_3_); 2.30–2.14 (m, 2H, CH_2_); 1.99–1.76 (m, 2H, CH_2_).

^13^C NMR (100 MHz, DMSO-d_6_) δ 194.3, 193.1, 153.9, 151.2, 148.7, 138.8, 136.8, 132.1, 130.5, 128.8, 125.6, 122.8, 120.0, 118.4, 112.5, 107.4, 106.3, 55.1, 36.41, 31.1, 21.9, 16.4.

The HRMS (ESI) was calculated for C_25_H_23_N_2_O_3_ [M + H]^+^ 399.17085; found 399.16998.

##### 7-benzoyl-11-phenyl-2,3,4,5,10,11-hexahydro-1H-dibenzo[b,e][1,4]diazepin-1-one (**4d**)

Compound **4d** (12.0 mg, 25%) was obtained from benzaldehyde (12.2 mL, 0.12 mmol) and 3-((2-amino-5-benzoylphenyl)amino)cyclohex-2-en-1-one (36.0 mg, 0.12 mmol) in the same manner as described for **FC2**, as a yellow powder.

^1^H NMR (400 MHz, DMSO-d_6_) δ 8.92 (s, 1H, NH); 7.71–7.29 (m, 11H, arom); 7.15–7.08 (m, 2H, arom); 6.69 (d, J = 8.4, 1H, CH); 5.79 (d, J = 6.4, 1H, NH); 2.75–2.57 (m, 2H, CH_2_); 2.31–2.16 (m, 2H, CH_2_); 2.01–1.79 (m, 2H, CH_2_).

^13^C NMR (100 MHz, DMSO-d_6_) δ 194.2, 193.1, 156.9, 148.4, 145.2, 144.6, 138.9, 136.1, 132.0, 130.3, 129.5, 128.8, 127.8, 125.6, 122.8, 120.0, 119.5, 118.4, 112.5, 112.2, 55.1, 36.4, 31.1, 21.8.

The HRMS (ESI) was calculated for C_26_H_23_N_2_O_2_ [M + H]^+^ 395.17594; found 395.17527.

##### 7-benzoyl-11-(p-tolyl)-2,3,4,5,10,11-hexahydro-1H-dibenzo[b,e][1,4]diazepin-1-one (**4e**)

Compound **4e** (7.0 mg, 17%) was obtained from 4-methylbenzaldehyde (12.0 mL, 0.1 mmol) and 3-((2-amino-5-benzoylphenyl)amino)cyclohex-2-en-1-one (30.5 mg, 0.1 mmol) in the same manner as described for **FC2**, as a yellow powder.

^1^H NMR (400 MHz, DMSO-d_6_) δ 8.86 (s, 1H, NH); 7.67–7.42 (m, 8H, arom); 7.28 (d, J = 8.4, 2H, arom); 7.16–7.09 (m, 2H, arom); 6.67 (d, J = 8.0, 1H, CH); 5.80 (d, J = 46.0, 1H, NH); 2.75–2.58 (m, 2H, CH_2_); 2.52 (s, 3H, CH_3_); 2.29–2.15 (m, 2H, CH_2_); 1.97–1.78 (m, 2H, CH_2_).

^13^C NMR (100 MHz, DMSO-d_6_) δ 194.3, 193.0, 156.7, 147.9, 145.3, 143.1, 138.8, 135.8, 132.4, 130.1, 129.6, 128.9, 126.6, 125.7, 122.5, 120.3, 120.1, 116.7, 112.4, 112.2, 55.1, 36.3, 31.1, 25.3, 21.8.

The HRMS (ESI) was calculated for C_27_H_25_N_2_O_2_ [M + H]^+^ 409.19159; found 409.19100.

Synthesis of **4f**–**v** can be found in Appendix A.

### 4.2. Cell Culturing and Cell Viability Assays

MDA-MB-231 cells (human breast adenocarcinoma cell line) were grown in standard conditions (37 °C, 5% CO_2_) in RPMI medium supplemented with 10% serum, 1 mg·mL^−1^ penicillin, 100 μg·mL^−1^ streptomycin and 2 mM l-glutamine. Cell viability was evaluated by using PrestoBlue^TM^ Cell Viability Reagent (Thermo Fisher Scientific, Rodano, Italy). Resazurin, the active nontoxic and non-fluorescent ingredient, is continuously converted in resorufin, a red fluorescent compound, in viable cells. Cells were seeded on black-wall, clear bottom 96-well microplates at a density of 50,000 cells/well. After 24 h, cells were treated with the FC2 derivatives from 1.5 μM to 100 μM (vehicle, DMSO) in the absence or presence of TNF at the concentration of 50 ng/mL. After 48 h, the fluorescence was determined using a fluorescence plate reader (Tristar2 S, Berthold Technologies, Bad Wildbad, Germany) equipped with 560 nm excitation and 585 nm emission filters. At least five replicates were performed for each condition. The EC_50_, was calculated plotting the percentage of cell survival against the compound concentration (A) and fitting the data with:$$\%\;\mathrm{survival}=\left(\mathrm{Tmax}-\mathrm{Tmin}\right)\frac{A}{A+\mathrm{EC}50}+\mathrm{Tmin}$$
where Tmax and Tmin are the maximum and minimum effect induced by the compounds. Analysis was performed with IgorPro 8.04 software (Lake Oswego, OR, USA).

### 4.3. Virtual Docking

The BIR1 domains (chain D, 3M0A [15] for cIAP2-BIR1 and chain B of 2POP [15,16] for XIAP-BIR1) and the FC2 or FC2 derivatives were prepared using the program Python Molecule Viewer 1.4.5 (http://mgltools.scripps.edu/packages/pmv accessed on 28 December 2022) to add hydrogen atoms and charges and set rotatable bonds for the ligands. The grid was centered on cIAP2-BIR1 Met33; grid sides: 28.5, 28.5, 22.5 Å. For XIAP-BIR1, a grid containing the entire domain was built (grid sides: 38.25, 42, 36 Å; spacing between nodes: 0.375 Å). AutoDock4 [26] was then used for the docking analysis, carrying 50 independent genetic algorithm cycles, and final results were analyzed with Python Molecular Viewer 1.4.5 [27]. In all the docking simulations, the protein molecules (cIAP2- and XIAP-BIR1) were constrained as a rigid body, while the ligands were allowed free rotation around all single bonds. 

### 4.4. Cloning, Expression and Purification of the BIR1 Domains of cIAP2 and XIAP

The human cIAP2/XIAP-BIR1 domains were cloned in pET28b and expressed and purified, as already described [11]. Briefly, plasmids for cIAP2-BIR1 or XIAP-BIR1 expression were used to transform Escherichia coli strain BL21-CodonPlus (DE3)-RP competent cells (Agilent). After induction with 0.5 mM isopropyl-β-D-thiogalactopyranoside (IPTG, Sigma-Aldrich), bacteria were grown for 3 h at 37 °C. After cell lysis, the expressed proteins were purified using Ni-NTA affinity column (His-trap FFcrude, Cytiva) and size exclusion chromatography (Superdex 75, Cytiva). Proteins were concentrated with Amicon Ultra centrifugal filters (with 3 kDa cut-off) in buffer 50 mM Tris-HCl pH 8.0, 200 mM NaCl, 10 mM DTT. Dynamic Light Scattering assays were performed in pUNK instrument (Unchained Labs) at a concentration of 1 mg/mL and confirmed the high quality of the samples obtained, revealing a sample size polydispersity lower than 20% for all protein constructs and hydrodynamic radii comparable to the monomeric forms of the proteins.

### 4.5. Thermal Stability Assays

Thermal denaturation assays were conducted in a StepOne Real Time PCR System (Applied biosystem Thermo Fisher Scientific, Waltham, MA, USA), with protein concentration of 40–50 μM and protein:ligand ratio 1:10. Samples were heated from 20 to 95 °C (0.2 °C/min) measuring fluorescence intensity at the Sypro orange excitation/emission ranges 470–505/540–700 nm, respectively.

### 4.6. Binding Assays

Trp fluorescence variation was used to determine the protein-ligand dissociation constant Kd at 23 °C in 50 mM Tris HCl, pH 8.0 containing 200 mM NaCl, 10% glycerol, 10 mM DTT. The fluorescence intensity emitted by XIAP-BIR1 Trp73 or cIAP2-BIR1 Trp76 was measured by varying the concentration of the different ligands. The tests were performed with a spectral fluorimeter (Varian Cary Eclipse Fluorescence Spectrophotometer, Agilent Technologies), recording the fluorescence emission spectra between 300 and 400 nm (excitation 280 nm). XIAP-/ cIAP2-BIR1 protein samples were concentrated at 5 μM.

Two-fold dilutions of FC2 and derivatives prepared to reach final concentrations ranging from 106 to 0.2 μM. 8 μL of each FC2 dilution were mixed with 180 μL of the protein solutions and fluorescence was measured in a 200 μL quartz cuvette. The Kd values were obtained through the GraFit5 program (Erithacus Software Limited, 2010), fitting the fluorescence values (*F*) with the following equation dependent on three parameters (*M*, *m*, *K_d_*):$$F=M-\frac{\left(M-m\right)}{\left[P_{T}\right]}\left[P_{I}\right];\;\mathrm{with}\;\left[P_{I}\right]=\frac{\left[P_{T}\right]+\left[I_{T}\right]+K_{d}-\sqrt{{\left(\left[P_{T}\right]+\left[I_{T}\right]+K_{d}\right)}^{2}-4\left[P_{T}\right]{\left[I_{T}\right]}^{2}}}{2}$$
where *F* is the fluorescence intensity, [*P_T_*]/[*I_T_*] are the total protein/inhibitor concentrations, *M*/*m* is the max/min of fluorescence and [*P_I_*] is the concentration of the protein bound to the inhibitor. All the analyses were performed with GraFit version 5 (Erithacus Software Ltd., Horley, UK).

### 4.7. Statistical Analysis

All results are expressed as means ± standard error of the mean (SEM) of at least three independent experiments.

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
