# Peer review of "New Class of Benzodiazepinone Derivatives as Pro-Death Agents Targeting BIR Domains in Cancer Cells"

_molecules, 2023, doi:10.3390/molecules28010446_

Round 1

Reviewer 1 Report

Very well presentation of the SAR study for FC2 derivatives and selection of the most relevant compound from the new series.

It is unclear how docking models will explain cell viability reduction by the FC2 derivatives.  

A discussion of the best binders with cIAP2- and XIAP-BIR1 should be included in the discussion section and the selection of the best compound is desirable.

Author Response

Very well presentation of the SAR study for FC2 derivatives and selection of the most relevant compound from the new series.

Q1.1) It is unclear how docking models will explain cell viability reduction by the FC2 derivatives. 

Answer 1.1) The virtual docking supports the structure-driven design, by (i) guiding the selection of FC2 modifications improving the affinity towards the target surface, (ii) describing the derivatives’ predicted binding on BIR1 surfaces and mapping gained/lost interactions with residues included in the target surface and, finally, (iii) identifying new putative hotspots/residues to be targeted by the derivatives. Direct explanation of cell viability reduction by virtual docking models is not claimed by this work. However, our objective is the design of derivatives of FC2, which demonstrated to target BIR1 surfaces involved in PPIs (protein-protein interactions) crucial for cancer cell survival, thus inducing cancer cell death. Improved binding of the derivatives vs the BIR1 domain should correlate with enhanced cancer cell death. For this reason, in the manuscript we hypothesize some correlations between predicted gained or lost interactions of BIR1/derivatives’ complexes, and the cytotoxic effect of the derivatives.

Action 1.1) In order to clarify the significance of virtual docking searches and avoid misunderstanding about the role of virtual docking we proceeded as follows:

  • at line 117 the sentence “This substitution negatively affected the potency (EC50 = 42.76 μM), suggesting that these hydrogen atoms may be involved in key H-bond interactions with the biological target or that reduced space exists for accommodating the methyl groups, as hypothesized in the structural analysis.“ was deleted;
  • a better explanation was introduced in Section 2.4 (amended manuscript), in line 192, anticipating the information previously reported in Section 2.4.1: “FC2 compound was identified in virtual screening against type I BIR domains of IAPs. In particular, the α1-turn-α2 region of BIR1 was set as target surface, since it is involved in protein-protein interactions (PPI) crucial for cancer cell survival. Virtual docking simulations of FC2 derivatives performed against the whole surface of BIR1 domains confirmed that FC2-derivatives preferentially target the same hotspot region (the α1-turn-α2 region) predicted for FC2 [14] (Figure 1, a), suggesting their ability to perturb the assembly of BIR1-mediated complexes between cIAP2/TRAF2 and XIAP/TAB1. FC2 derivatives also show free energies of binding comparable to FC2, with estimated inhibition constants (Ki) in the low micromolar range (Table 2). Major deviations were calculated for 4h in cIAP2-BIR1 and for 4g in XIAP, whose predicted Ki reached 9.09 µM and 4.83 µM, respectively. Virtual docking simulations allowed to describe the derivatives’ interactions with the target proteins and to hypothesize correlations between some predicted hotspot interactions and observed enhancements in cytotoxicity”.

Q2.1) A discussion of the best binders with cIAP2- and XIAP-BIR1 should be included in the discussion section and the selection of the best compound is desirable.

Answer 2.1) We are thankful for the suggestion as the determination of the derivatives’ affinity for cIAP2- and XIAP-BIR1 is relevant to the objective of the work.

Action 2.1) In the discussion section the paragraph at line 376 “The affinity of the best binders is comparable to the one observed for the progenitor FC2, in the low micromolar range.” was substituted with “According to experimental results, FC2 retains the best measured Kd. Among novel FC2-derivatives, compounds 4a, 4d, 4g, 4h, 4j, 4m, 4o, 4p, 4q are the best binders, displaying Kd values in the low micromolar range. Such results generally confirm beneficial effects of electron withdrawing groups addition in R1 in the meta position of the phenyl ring”.

Author Response

Q 1.2) In molecular simulation section, authors have reported only molecular docking results.

Molecular docking results should be properly validated by running molecular dynamics simulation studies. Authors can make a note on this.

Answer 1.2) We are thankful for the suggestion and we understand the significance of molecular dynamics in this context, in order to analyse the putative binding of FC2-derivatives upon movements of the BIR1 homologous structures. We are planning to perform such experiments in a more detailed study focusing on the best candidates, selected after in vitro screening on cancer cells and in vitro characterization on purified proteins, and combining these simulations with experimental data.

Action 1.2) To point out the importance of MD approach, we added a sentence in the discussion section “Combination of molecular dynamics and other experimental structure-based techniques (i.e. crystallography, NMR) are in progress to better characterize the molecular details of BIR1 binding with selected compounds, 4m and 4p.” at line 387.

Q 2.2) Abstract to be written more precise.

Answer 2.2) Details about the ambiguous part of the abstract are missing. Nevertheless, we noticed that techniques where not properly described. The word count was then set to 199 (200 maximum).

Action 2.2) We added a sentence in the abstract about methods used, at line 25: “Cytotoxic effect of FC2-derivatives was evaluated in MDA-MB-231 and binding to the cIAP2- and XIAP-BIR1 domains was assessed in fluorescence-based techniques and virtual docking.”

Q 3.2) Reason for the specific use of MDA-MB-231 cell lines to be explained in a rational way.

Answer 3.2) As reported in 2.2.1, the procedures adopted in the cell-based assays are the ones reported in Cossu et al [14], where the progenitor compound FC2 was identified and characterized. Such procedures include the use of MDA-MB-231 cell line (sensitive to FC2 as single agent) for the screening of FC2 analogues and for combination treatments with 50 ng/ml of TNF. At the concentration of 50 ng/ml, in MDA-MB-231 TNF is not toxic, and ensured triggering of NF-κB, that is essential to evaluate possible enhancements of FC2-derivatives cytotoxicity and the role of FC2-derivatives in the modulation of NF-κB. The use of MDA-MB-231 here reported is a continuation of previously reported work, which we plan to expand, investigating other cell lines, where the IAPs play major role in cancer progression and metastasis. In fact, this cancer cell line has been extensively investigated in the study of IAP-targeting therapies.

Action 3.2) The beginning of Section 2.2.1, line 108, was modified as follows: “The screening of FC2-like molecules initiated by Cossu et al. [14] was performed in a human breast adenocarcinoma cells (MDA-MB-231). Such a cell-line is extensively investigated in the presence of IAP-targeting compounds and was used to characterize the mechanism of FC2-induced cell death and regulation of NF-κB [14]. For this reason, MDA-MB-231 cells were adopted to continue the screening of novel FC2-derivatives (Table 1).”

Q 4.2) It is very useful to the readers, if the mechanism of chemical reactions in each transformation could be presented in the manuscript.

Answer 4.2) We are thankful for the suggestion. In order to facilitate the readers, we elaborated a scheme summarizing the plausible mechanism for the formation of compound 4d, as reference derivative.

Action 4.2) We added Scheme S1 in the supplementary material and a sentence in section 2.1, line 101 of the manuscript, as follows: “Mechanism hypothesis for the formation of 1H-dibenzo[1,4]diazepin-1-one derivatives is summarized in Scheme S1.”

Q 5.2) Material methods to be written concisely.

Answer 5.2) In our opinion the Material and Methods section reports the essential details instrumental to reproduce the experiments.

            Action 5.2) No action was taken.

Reviewer 3 Report

This manuscript by Fiore et al. describes “New class of benzodiazepinone derivatives as pro-apoptotic agents targeting BIR domains”. Authors synthesized twenty-two new derivatives of FC2 and evaluated these compounds for their cell viability against MDA-MB-231 cancer cells. Some compounds displayed good EC50 in the absence of TNF and a few displayed increased activities in combinations with TNF. Authors did virtual docking simulations of these derivatives against type I BIRs, performed thermal stability studies, and binding affinity using fluorimetric assays. This manuscript is well written with clear discussions. I would recommend this paper to be considered after minor points

1.   Chemical structure can be reduced in size so that FC2 is visible in table 1.

2.  Were there any reasons authors selected 50 ng/mL of TNF for combination studies with these derivatives? It could have been lower than 50 ng/mL.

3. Please upload supplementary information with full characterization data (proton and carbon NMRs) of all compounds. I think there should be 28 carbon signals in compound 4a. Similarly, there should be 22 carbon signals for compound 4b. Authors should recheck the experimental data, and missing signals can be added in revised manuscript. HRMS formula for compound 4d should be corrected for M+H found 409,117529.

Author Response

This manuscript by Fiore et al. describes “New class of benzodiazepinone derivatives as pro-apoptotic agents targeting BIR domains”. Authors synthesized twenty-two new derivatives of FC2 and evaluated these compounds for their cell viability against MDA-MB-231 cancer cells. Some compounds displayed good EC50 in the absence of TNF and a few displayed increased activities in combinations with TNF. Authors did virtual docking simulations of these derivatives against type I BIRs, performed thermal stability studies, and binding affinity using fluorimetric assays. This manuscript is well written with clear discussions. I would recommend this paper to be considered after minor points

Q1.3) Chemical structure can be reduced in size so that FC2 is visible in table 1.

Answer 1.3) The conversion to PDF format impaired visualization of the FC2 chemical structure in table 1. However, in the amended manuscript, we deleted the chemical structure of FC2 from the Table 1 since it is already reported in Scheme I.

Action 1.3) Since the structure of FC2 is reported also in Scheme I, we deleted the structure in Table 1, and added a sentence in the Table 1 legend “The position of R and R1 substitutions is reported in Scheme I.” The files were updated accordingly.

Q 2.3) Were there any reasons authors selected 50 ng/mL of TNF for combination studies with these derivatives? It could have been lower than 50 ng/mL.

Answer 2.3) Please see Answer 3.2.

Action 2.3) At line 180, we stressed the connection with previous work (reference 14), with the sentence as follows: “Since the cytokine TNF stimulates the TNF-R and the recruitment of all molecular actors (i.e. IAPs) triggering the NF-κB pathway, the effect of each compound on cell viability was measured both as a single agent and in combination with 50 ng/ml of TNF (Table 1), as reported by Cossu et al. [14], where induction of cell death with FC2 was shown to be potentiated by co-treatment with TNF.”

Q 3.3) Please upload supplementary information with full characterization data (proton and carbon NMRs) of all compounds. I think there should be 28 carbon signals in compound 4a. Similarly, there should be 22 carbon signals for compound 4b. Authors should recheck the experimental data, and missing signals can be added in revised manuscript. HRMS formula for compound 4d should be corrected for M+H found 409,117529.

Answer 3.3) The number of signals in the 13C spectra are sometimes lower than the number of carbon atoms present in the molecule for several reasons. In our case, a coalescence or overlapping of some signals is possible, especially in the aromatic part.

Action 3.3) According to the aforementioned explanation, no action was taken. Please note that we corrected in the Material and Methods, line 510, “HRMS (ESI) calculated for C26H23N2O2 [M+H]+ 395.17594; found 409,117529” to “395,17527.”
